# Cost-Effective Method for Full-Length Sequencing of Monoclonal Antibodies from Hybridoma Cells

**DOI:** 10.3390/antib14030072

**Published:** 2025-08-22

**Authors:** Sarah Döring, Georg Tscheuschner, Sabine Flemig, Michael G. Weller, Zoltán Konthur

**Affiliations:** Federal Institute for Materials Research and Testing (BAM), Richard-Willstätter-Strasse 11, 12489 Berlin, Germany; sarah.doering@bam.de (S.D.); georg.tscheuschner@bam.de (G.T.); sabine.flemig@bam.de (S.F.); michael.weller@bam.de (M.G.W.)

**Keywords:** full-length antibody sequencing, hybridoma cell loss, MALDI-TOF MS, immunoglobulin isotyping, RNA Illumina sequencing

## Abstract

Background: Monoclonal antibodies play an important role in therapeutic and analytical applications. For recombinant expression, the coding sequences of the variable regions of the heavy and light chains are required. In addition, cloning antibody sequences, including constant regions, reduces the impact of hybridoma cell loss and ensures preservation of the naturally occurring full antibody sequence. Method: We combined amplification of IgG antibody variable regions from hybridoma mRNA with an advanced method for full-length cloning of monoclonal antibodies in a simple two-step workflow. Following Sanger sequencing and evaluation of consensus sequences, the best matching variable, diversity, and joining (V-(D-)J) gene segments were identified according to identity scores from IgBLAST reference sequences. Simultaneously, the mouse IgG subclass was determined at the DNA level based on isotype-specific sequence patterns in the C_H_1 domain. Knowing the DNA sequence of V-(D-)J recombination responsible for the complementary determining region 3 (CDR 3), variable region-specific primers were designed and used to amplify the corresponding antibody constant regions. Results: To verify the approach, we applied it to the hybridoma clone BAM-CCMV-29-81 and obtained identical full-length antibody sequences as with RNA Illumina sequencing. Further validation at the protein level using an established MALDI-TOF MS-fingerprinting protocol showed that five out of six genetically encoded CDR domains of the monoclonal antibody BAM-CCMV-29-81 could be efficiently correlated. Conclusion: This simple, streamlined method enables the cost-effective determination of the full-length sequence of monoclonal antibodies from hybridoma cell lines, with the added benefit of obtaining the DNA sequence of the antibody ready for recombinant expression.

## 1. Introduction

With their breakthrough in hybridoma technology, Köhler and Milstein revolutionized the production of monoclonal antibodies by enabling their unlimited supply [1]. By immunization with defined biomolecules, antibodies can be produced for various therapeutic, diagnostic, or research applications. Antibody diversity results from somatic recombination in B lymphocytes, where V (variable)-, D (diversity)-, and J (joining)-gene segments are rearranged [2]. The D segment is exclusive to the heavy chain (V_H_), whereas variable regions of light chains (V_L_) are generated from V- and J-gene segments alone. This process creates hypervariable regions in the antibody paratope, known as complementarity-determining regions (CDRs), which are antigen-specific and structurally essential for target recognition and interaction [3]. Through this evolutionary process, a large variety of selective and high-affinity monoclonal antibodies can be generated despite the limited number of genes.

The production of monoclonal antibodies in hybridoma cells requires several weeks of cultivation, and some clones yield only small amounts of antibody. Recombinant expression of antibodies in mammalian cells is therefore increasingly preferred, as it can yield large quantities in a shorter time under much better-controlled conditions [4]. To generate expression plasmids, knowledge of the sequence of the antibody heavy and light chain variable regions is essential. To date, several techniques are available for sequencing of variable region or full-length antibodies derived from hybridoma cells, B lymphocytes, or phage display in vitro selection protocols. These include next-generation technologies such as RNA Illumina [5,6] or Oxford Nanopore sequencing [7,8]. Both methods require expertise for library preparation and bioinformatic analysis. Furthermore, sending samples to commercial sequencing services is expensive, costing hundreds or even thousands of dollars and taking several weeks to process [9]. Although useful for sequence identification, the approach does not eliminate the need for cloning the heavy and light chains when recombinant expression is intended. In most cases, however, RNA Illumina sequencing is still the method of choice for high-throughput antibody sequencing because it allows sequencing of multiple samples in parallel [10]. More recently, full-length single-cell B cell transcriptomics paired with haplotype-resolved germline immunoglobulin references can read complete heavy and light chain transcripts and assign isotypes unambiguously [11], but these platforms remain specialized and computationally intensive for routine use. In parallel, fluorescence-activated droplet sequencing can directly sequence rare hit droplets and recover paired chains without physical sorting [12], although it requires dedicated microfluidic hardware.

Protein-based antibody sequencing is another alternative, using advanced tandem mass spectrometry to reconstruct full-length antibody sequences *de novo* at the protein level. High-resolution instruments such as Orbitrap or Q-TOF mass spectrometers are typically used to generate overlapping peptide fragments by multienzyme digest that are computationally assembled into the complete antibody sequence [13,14]. While these approaches can provide valuable insights, including the identification of post-translational modifications [15], they are generally time-consuming, technically demanding, and cost several thousand dollars per antibody [16,17], making them impractical for routine applications. Comprehensive reviews highlight a broader trend toward integrating single-cell sequencing, proteomics, and artificial intelligence for discovery, but with substantial cost and complexity for everyday laboratory workflows [18].

In contrast, the low-throughput Sanger technology used in this work allows the antibody to be sequenced using PCR-based methods [19,20,21,22]. Advantageously, continuous read lengths of up to 1000 bp are possible and do not require computer-assisted assembly of many short fragments as in RNA Illumina sequencing. Using 5′ RACE-SMART technology, chain-specific primers bind to conserved regions in the constant domains of mouse or human IgG heavy and light chains (kappa or lambda) [21]. With this protocol, the variable sequence of the antibodies can be determined in-house in a few days at a low cost. The approach is convincing due to its simplicity, robust data analysis, and particular suitability for sequencing small numbers of antibodies [23,24,25]. Furthermore, this method can be used to analyze regulatory elements within the untranslated 5′ region (5′ UTR) of the RNA, in addition to the leader sequences of the heavy and light chains of the antibody. These elements have the potential to influence mRNA transcription and antibody production [26].

Although knowledge of the antigen-binding region is sufficient for diagnostic antibody production, the complete sequence should be investigated when using monoclonal antibodies from hybridoma cells. The IgG constant region can also influence how strongly and specifically an antibody binds to its target [27]. Differences in the constant region, for example, between subclasses, can change the position or flexibility of the antigen-binding sites and thereby alter binding properties. In recent years, several experiments from different publications could not be reproduced due to quality problems with monoclonal antibodies [28,29,30]. Therefore, providing the amino acid sequence would be a first step towards improving the quality of research. Moreover, the storage of full-length sequence information on DNA plasmids avoids expensive cryo-storage of cell lines, hybridoma clone losses, and unproductivity due to mutations or external influences [31,32,33].

To address these needs, we combined the simple method of sequencing the variable regions of mouse IgG antibodies from hybridoma cell lines [21] with further steps to enable cost-effective sequencing and cloning of full-length antibodies (Figure 1). As a practical example, we sequenced a recently published mouse monoclonal IgG antibody derived from a hybridoma cell line (BAM-CCMV-29-81) against the Cowpea Chlorotic Mottle Virus (CCMV) [34]. In the first step of the workflow, RNA extracted from hybridoma cells was used for RT-PCR amplification of the antibody variable regions and cloning into sequencing vectors. After Sanger sequencing of several clones and evaluation of consensus sequences, the best-matched V-(D-)J gene segments based on identity scores were successfully pinpointed using IgBLAST reference sequences. Because IgG2a and IgG2c are difficult to distinguish at the protein level [35,36,37], we identified subclass-specific motifs in the C_H_1 exon and assigned the isotype at the DNA level, consistent with prior reports [38]. Using the V-(D-)J sequences, we then designed CDR3-anchored primers to amplify the constant regions of the heavy and κ-light chains, linking variable and constant domains in one step. To confirm the specific CDR sequences at the protein level, we applied a rapid MALDI-TOF-MS fingerprinting protocol [39]. This method additionally allows rapid verification of clone identity and subclass, thereby supporting quality control during antibody production in hybridoma cells. Finally, we demonstrated that our workflow provides full-length antibody sequences equivalent to those obtained by RNA Illumina sequencing. The complete process can be completed in less than two weeks and with lower cost and less complexity than commercial full-length services. In addition, the obtained DNA material can be used to generate recombinant expression plasmids without requiring commercial DNA synthesis, including the natural secretion leader peptide.

## 2. Materials and Methods

### 2.1. Biochemicals and Hybridoma Production

Acetonitrile 99.95% (ACN, 2697), acetic acid 99.5% (2234), agarose (9920-500G), LB agar (8843-500G), LB medium Miller (8822-500G), and nuclease-free water (7711) were purchased from Th. Geyer GmbH & Co. KG (Renningen, Germany). Ampicillin sodium salt ≥ 97% (K029.4) was obtained from Carl Roth (Karlsruhe, Germany). Bruker Peptide Calibration Standard II (8222570) was purchased from Bruker (Billerica, MA, USA). Tris(2-carboxyethyl) phosphine hydrochloride 98% (TCEP, 580560) was obtained from Calbiochem (San Diego, CA, USA). 2,5-Dihydroxyacetophenone (DHAP, A12185) was purchased from Alfa Aesar (Haverhill, MA, USA). EDTA-Disodium salt ≥ 97% (03685), sulfuric acid (5 mM, 28-6020), sinapinic acid > 98% (D7927), tris-(hydroxymethyl)-aminomethane hydrochloride (T3253), tris(hydroxymethyl)aminomethane (T1503), and recombinant trypsin (3708985001) were purchased from Sigma-Aldrich (St. Louis, MO, USA). Trifluoroacetic acid 99.5% (TFA, 85183) and Pierce C18 tips (10 µL, 87784) were obtained from Thermo Fisher (Waltham, MA, USA). Phosphate-Buffered Saline (PBS, L1825) was purchased from Biochrome GmbH (Berlin, Germany). Tris(hydroxymethyl)-aminomethane (108382) and glucose monohydrate (1083422) were obtained from Merck KGaA (Darmstadt, Germany). Deoxynucleotide (dNTP) Solution Mix (N0447L), NEB^®^ 10-beta Competent *E. coli* (C3019H), NEB 10-beta/Stable Outgrowth Medium (B9035S), and RNAse inhibitor (M0314S) were purchased from New England Biolabs (Ipswich, MA, USA). Lab water was used from a Milli-Q water purification system of Millipore (Bedford, MA, USA) with a resistivity of >18.2 Ω and a TOC value of <5 ppb. Primers (Eurofins Genomics, Ebersberg, Germany) and corresponding oligonucleotide sequences are in Appendix A, Appendix A.

The mouse anti-CCMV monoclonal antibody clone BAM-CCMV-29-81 was derived from an in-house antibody project [34]. Mouse immunization and hybridoma cell production were performed by Davids Biotechnology GmbH (Regensburg, Germany).

### 2.2. RNA Extraction of Cryo-Preserved Hybridoma Cells

Preparation of cryo-preserved hybridoma cells for RNA extraction was based on the protocol of Andrew et al., 2019 [40]. Briefly, cryo-preserved vials containing 5 × 10^6^ hybridoma cells were thawed in a 37 °C water bath for 5 min and transferred to a 1.5 mL reaction tube. After centrifugation at 500× *g* for 5 min, the supernatant was removed, and the cells were resuspended in 1 mL of ice-cold sterile PBS. The cells were centrifuged again as described, the supernatant was removed, and RNA extraction of the pelleted hybridoma cells was performed with TRIzol reagent (Invitrogen, Carlsbad, CA, USA, 15596026) according to the manufacturer’s instructions. RNA was further purified using the PureLink RNA Mini Kit (Invitrogen, Carlsbad, CA, USA, 12183020) according to the manufacturer’s instructions with additional on-column DNase digestion (Thermo Fisher, 12185010). Determination of RNA concentration and testing for protein or phenol contamination was performed on a NanoDrop One C (Thermo Fisher).

### 2.3. RT-PCR Amplification of Variable Regions

The RT-PCR protocol for cDNA synthesis and amplification of antibody variable regions, as well as the primers for transcription and PCR amplification, were based on the protocol of Meyer et al. 2019 [21]. Briefly, three cDNA synthesis reactions were prepared in PCR tubes for kappa (ĸ), lambda (λ), and heavy chain. Each PCR tube contained 2 μL of 50 ng/μL RNA, 1 μL of 10 μM reverse RT primer for a specific antibody chain (Appendix A), and 1 μL of 10 mM dNTPs. In addition, a master mix containing the remaining reagents was prepared for all three cDNA synthesis reactions: 5.1 μL nuclease-free water, 6 μL 5x SMARTScribe buffer, 3 μL 20 mM DTT, and 0.9 μL 100 μM template-switch oligonucleotide (Appendix A). While PCR tubes containing RNA were incubated at 72 °C for 3 min to denature RNA secondary structures, 1.5 μL 100 U/μL SMARTScribe Reverse Transcriptase (#639537, Takara Bio, San Jose, CA, USA) and 0.75 μL 80 U/μL RNAse inhibitor were added to the master mix. Then, 6 μL of the master mix was added to each PCR tube and incubated at 42 °C for 60 min, followed by 70 °C for 5 min to stop the reaction and cooling at 4 °C.

The cDNA generated was then used for PCR amplification with Phusion Hot Start II DNA-Polymerase (F549L, Thermo Fisher), and reactions for V_H_ and V_L_ chains were carried out in two independent tubes. Each reaction contained 10 μL 5x Phusion HF buffer, 1 μL 10 mM dNTPs, 3 μL cDNA from the RT reaction, 2.5 μL 10 μM universal forward primer ISPCR, 2.5 μL 10 μM chain-specific reverse PCR primer, 30.5 μL nuclease-free water, and 0.5 μL 2 U/μL Phusion polymerase. The step-down PCR conditions were as follows: initial denaturation step at 98 °C for 30 s; 10 cycles of 98 °C for 15 s, primer hybridization at 63 to 57.5 °C for 30 s with decreasing temperature by 0.5 °C with each cycle, and extension at 72 °C for 30 s; 15 cycles of 98 °C for 15 s, 56 °C for 30 s, and 72 °C for 30 s; followed by final extension 72 °C for 7 min. Amplicons were analyzed in a 1% agarose gel with 1x TAE running buffer at 90 V for 30 min. The GeneRuler^TM^ 100 bp Plus DNA Ladder was used as a standard. To prepare PCR products for cloning, the complete PCR reaction was then run again in agarose gel electrophoresis under the above conditions. PCR products corresponding to the amplified variable heavy and light chain regions were extracted from the gel using the E.Z.N.A.^®^ MicroElute^®^ Gel Extraction Kit (Omega Bio-tek Inc., Norcross, GA, USA).

### 2.4. RT-3′RACE-PCR Amplification of Constant Regions

In general, reverse transcription and PCR amplification of cDNA were performed as described in Section 2.3. However, reverse transcription of RNA to cDNA was performed using an AOligo(dT)-ISPCR primer (Appendix A) with an adaptor sequence that binds to the 5′ end of the poly-A signal. In addition, CDR-specific forward primers for heavy (CCMV-mIGHG_VDJ-PCR) and light (CCMV-mIGK_VJ-PCR) chains were used in combination with the universal forward primer ISPCR (Appendix A), which binds to the adaptor sequence, for antibody-specific PCR amplification of the constant regions. The amplicons were analyzed and extracted from the agarose gel for cloning as described in Section 2.3.

### 2.5. Vector Cloning, Sanger Sequencing, and Data Evaluation

Blunt-end ligation of purified PCR products was performed using the CloneJET PCR Cloning Kit (Thermo Fisher, K1232) according to the manufacturer’s protocol. A total of 2 μL of each reaction was transformed into chemically competent *E. coli* NEB^®^ 10-beta cells via heat shock. A total of 100 μL of each transformation was plated onto LB agar plates containing 100 μg/mL ampicillin and 2% glucose and incubated overnight at 37 °C. A total of 20 to 25 grown colonies per antibody chain were analyzed by colony PCR to exclude false-positive clones. Amplification of the inserted antibody chains was performed with the vector-specific primer pair pJET1.2 forward and pJET1.2 Reverse (Appendix A) using OneTaq^®^ DNA Polymerase (NEB, M0480). Each 15 µL volume reaction contains 3 µL of 5x OneTaq Standard Reaction Buffer, 0.3 µL of 10 µM of each primer, 0.3 µL of 10 mM dNTP and 0.375 U of OneTaq DNA polymerase. The amplification conditions were an initial denaturation at 94 °C for 5 min, followed by 30 cycles of 94 °C for 15 s, 60 °C for 30 s, and 68 °C for 1 min, and a final extension step of 68 °C for 5 min. PCR products were analyzed in 1% agarose gel as described above. Cloned plasmid constructs that produced a PCR product greater than 750 bp for the V_H_ and V_L_ regions and 1200 bp (C_H_) and 700 bp (C_L_) constant chains were declared as positive clones.

Positive clones were inoculated into 5 mL LB medium supplemented with 100 μg/mL ampicillin and 2% glucose and grown overnight at 37 °C with shaking at 200 rpm. Bacterial cells were harvested by centrifugation, and plasmid DNA was isolated from the cell pellet using the E.Z.N.A.^®^ Plasmid DNA Mini Kit I. DNA samples were sent to LGC Genomics GmbH for overnight Sanger sequencing using the pJET1.2 Forward and/or pJET1.2 Reverse sequencing primers. The sequencing data for each clone of a given antibody chain were aligned using the Clustal Omega (version 1.2.4) multiple sequence alignment tool available at EMBL-EBI (https://www.ebi.ac.uk/jdispatcher/msa/clustalo), last accessed on 20 August 2025 [41] to assess sequence consensus. These sequences were then compared with constant region sequences retrieved from the IMGT database (Section 2.6). For the variable heavy (VH) and variable light (VL) chains, the DNA sequences were further compared to IgBLAST reference sequences using default parameters and mouse as the query organism to determine percentage identity (https://www.ncbi.nlm.nih.gov/igblast/), last accessed on 20 August 2025. The nucleotide sequences for the heavy and light chains of monoclonal anti-CCMV monoclonal antibody clone BAM-CCMV-29-81 have been deposited in GenBank under accession numbers [PX123807] and [PX123808], respectively.

### 2.6. Bioinformatic Analysis of Constant Mouse IgG Sequences

The IMGT database was initially queried for all protein-coding regions of the constant heavy chains for all available IgG subclasses, as well as the constant regions of κ- and λ-light chains. The data were based on genome assemblies for *Mus musculus* strains: C57BL/6J (as of February 2024; NC_000078.7, NC_000072.7, NC_000082.7), BALB/c (as of June 2020; NT_096355.1), and NOD (as of July 2016; Y10606.1). Sequence alignments were performed using Clustal Omega [41], comparing the respective exons of the constant regions of the heavy light chain variants. The highly conserved, yet subclass-specific, sequences found in the C_H_1 region of the mouse IgG subclasses were used to enable isotype identification at the DNA level. A similar approach was applied to differentiate between the various lambda light chains. To determine the primer binding sites, the respective oligonucleotide sequences were converted into their reverse complements and analyzed for potential mismatches in primer annealing.

### 2.7. Determination of Mouse IgG Isotype on Protein Level

The subclass and light chain variant of the mouse monoclonal antibody of clone BAM-CCMV-29-81 were determined using the IsoStrip^TM^ Mouse Monoclonal Antibody Isotyping Kit from Roche (Basel, Switzerland) according to the manufacturer’s instructions. Briefly, 150 µL of purified antibody (36 µg/mL) was added to the development tube and incubated for 30 s. After homogenization, the isotyping stripes were added to the development tube for 1 min. Results were interpreted after a further 10 min.

In addition, the Mouse Immunoglobulins Isotyping Universal Module (MQUM1) with specific Mouse-Isotyping Buffers (MQSBG1, MQSBG2A, MQSBG2B, MQSBG2C, MQSBG3) and light chain variant buffers for lambda (MQSBLLC) and kappa (MQSBKLC) from Milenia Biotec GmbH (Gießen, Germany) were used. The purified antibody (36 µg/mL) and undiluted supernatant of cell line BAM-CCMV-29-81 cultures were tested. In a non-binding ELISA plate, 45 µL of the anti-mouse subclass or light chain sample buffers were added to individual wells and incubated with 10 µL of the antibody sample after a brief shaking. Dipsticks were then immersed in the appropriate solutions and incubated for five minutes.

### 2.8. MALDI-TOF-MS-Based Antibody Fingerprinting

#### 2.8.1. Cleavage with Diluted Sulfuric Acid

A 150 µL reaction mixture containing 10 µg antibody (stock solution 0.33 mg/mL), 5 mM sulfuric acid, and 0.1 M TCEP concentration was prepared in a 0.2 mL PCR tube and placed in an Eppendorf Thermomixer C (equipped with a SmartBlock PCR 96, Eppendorf SE, Hamburg, Germany) for 30 min at 99 °C and 950 rpm shaking to cleave the antibody. An alkylation step was omitted. Peptides were then enriched and washed using Pierce C18 tips (10 µL) according to the manufacturer’s protocol and eluted with 2 µL of 2,5-dihydroxyacetophenone (DHAP) MALDI matrix solution (10 mg/mL, 69.9% purified water, 30% ACN, 0.1% TFA) directly onto the MALDI target. MALDI measurements were performed on a Bruker Autoflex maX in reflector mode. The instrument was calibrated with the Bruker Peptide Calibration Standard II using DHAP as the MALDI matrix. A fingerprint spectrum was obtained by accumulating a total of 5000 laser shots.

#### 2.8.2. Enzymatic Cleavage with Trypsin

A 50 µL reaction mixture containing 10 µg antibody (stock solution 0.33 mg/mL), 0.1 M Tris buffer (pH 7.8), and 0.1 M TCEP concentration was prepared in a 0.2 mL PCR tube and placed in an Eppendorf Thermomixer C (equipped with a SmartBlock PCR 96) for 15 min at 99 °C and 950 rpm shaking to cleave disulfide bonds and denature the antibody. The reaction mixture was then cooled down to 55 °C, and trypsin was added at a trypsin-to-antibody mass ratio of 1:120. The tryptic digestion was continued at 55 °C for a further 15 min, followed by the addition of 10 µL of 0.1% TFA. Peptide purification and MALDI measurements were performed as described above.

### 2.9. RNA Illumina Sequencing

Hybridoma cells of clone BAM-CCMV-29-81 were sent to Absolute Antibody Ltd. (Oxford, UK) for whole transcriptome shotgun sequencing (RNA-Seq). Two vials of cryo-preserved cells containing 1 × 10^6^ hybridoma cells were thawed for 5 min in a 37 °C water bath and transferred to a 1.5 mL reaction tube. After centrifugation at 500× *g* for 5 min, the supernatant was removed, and the cells were resuspended with 1 mL of Preservation Buffer from the proprietary SeqPack Hybridoma Preservation Kit (Absolute Antibody Ltd., Oxford, UK) for shipment at room temperature. Upon receipt of the cells, total RNA was extracted, and a barcoded cDNA library was generated by random hexamer RT-PCR. The library was then sequenced using a high-throughput Illumina HiSeq sequencer (Illumina, San Diego, CA, USA). The variable heavy and light chain domains were identified separately, and the abundance of each gene was quantified in transcripts per million (TPM). Additionally, the species and isotype of the antibody genes were confirmed based on the constant domain sequences. To ensure data quality, Absolute Antibody Ltd. compared the identified sequences with known non-functional antibody genes commonly found in hybridomas, allowing for the exclusion of aberrant chains from further analysis.

## 3. Results

### 3.1. Evaluation of Primer Binding Sites

We first examined the binding sites of published primers [21] used for sequencing the variable region of the heavy, κ- and λ-light chains of mouse antibodies. For the heavy chain subtypes, we observed several mismatches in primer binding sites (Figure 2). The primer used for heavy chain–specific reverse transcription showed 100% homology with the IgG1, IgG2a, IgG2b, and IgG2c subclasses. However, five mismatches were identified in the IgG3 gene sequence, which may reduce reverse-transcription efficiency. In contrast, the primer for PCR amplification of the heavy chain variable region exhibited 100% identity for IgG1, IgG2a, and IgG2c. A single mismatch was identified in the conserved IgG2b sequence, which is not expected to affect amplification efficiency significantly. Additionally, a four-nucleotide mismatch was identified at the primer binding site of the IgG3 sequence used for PCR amplification.

For the κ-light chain, a single conserved constant region serves as the primer-binding site for reverse transcription and PCR amplification. The primer sequences show 100% identity to this region. In contrast, the mouse genome contains four different functional λ-light chain genes, and the primers show one to three mismatches within the binding sites of the primers (Appendix A). The primers used for reverse transcription and PCR amplification target the most conserved site available within the constant IgLC1 region. Together, these patterns suggest that amplification of IgG3 and some λ-light chains may be less efficient than κ-light chains and other IgG subclasses.

### 3.2. Sequencing of Variable Region and V-(D-)J-Recombination Analysis

For sequencing of the variable regions of the mouse monoclonal antibody derived from the hybridoma clone BAM-CCMV-29-81, RNA was extracted and amplified with heavy as well as ĸ- and λ-chain specific primers (Appendix A) by RT-PCR. The resulting amplicons were between 550 and 600 bp in size and indicated that the mouse anti-CCMV monoclonal antibody exclusively contains ĸ-light chain (Figure 3A).

After cloning into the vector, eight plasmids isolated from colony PCR-positive clones containing the variable region of each chain were sent for Sanger sequencing. All plasmids containing heavy chain sequences shared identical consensus sequences. They were used for IgBLAST analysis to verify the percent identity of variable regions to IgBLAST reference sequences and to identify the top-matching V-(D-)J genes (Table 1). Appendix A shows the complete IgBLAST report of the variable heavy chain query for the mouse monoclonal antibody of clone BAM-CCMV-29-81 with in-frame V-(D-)J-rearrangement and functional gene translation. In contrast, sequencing of κ-light chain plasmids revealed two variants. Both DNA sequences were analyzed with IgBLAST and revealed that one transcript contains a frameshift mutation in the V-J-gene junction, resulting in an early stop codon (Appendix A). This non-functional chain was identified in 50% of the ĸ-light chain samples sent for sequencing. However, the second identified variable region of ĸ-light chain shows no early termination codon (Appendix A). The identified V- and J-genes of the variable region from the productive κ-light chain are listed in Table 1. Knowing the specific DNA sequence of V-(D-)J recombination, we designed a hypervariable region-specific primer in the segment of the CDR3-domain for RT-PCR amplification of the associated constant heavy and ĸ-light chains (Section 3.4).

### 3.3. Identification of IgG Mouse Isotype at the DNA and Protein Level

Accurate determination of the mouse antibody isotype is particularly important in diagnostic applications such as ELISA or lateral flow assays, as it influences secondary antibody selection and assay design. Additionally, the IgG isotype impacts the efficacy of antibody purification using Protein-A or Protein-G affinity columns. To determine the IgG isotype of the monoclonal mouse antibody during variable region sequencing, we closely examined the conserved constant-region sequence in the C_H_1 exon. Accordingly, the C_H_1 exon reference sequences of IgG1, IgG2a, IgG2b, and IgG3 mouse antibodies typically produced by BALB/c or Swiss Webster mice and of IgG2c generated by C57BL/6J or NOD strains were obtained from the IMGT and NCBI databases (Appendix A). The alignment of constant C_H_1 reference sequences revealed the presence of short and highly conserved regions for each mouse IgG isotype, which are specific to their respective subclasses (Figure 2). Because the primers bind in the C_H_1 region of the antibody sequence, the RT-PCR product automatically contains these subclass-specific motifs. As a result, these unique sequence motifs effectively differentiate the individual subtypes following sequencing of the variable region of a monoclonal mouse IgG antibody.

The C_H_1 reference sequences of different mouse IgGs were aligned against the consensus sequence derived from the variable region of the mouse monoclonal antibody heavy chain of clone BAM-CCMV-29-81 (Figure 4C). Based on the unique C_H_1 exon motif that distinguishes mouse IgG subclasses, the BAM-CCMV-29-81 monoclonal antibody was clearly classified as IgG2c at the DNA level.

For protein-level confirmation, two different lateral flow isotyping kits were used for subclass determination. The first kit combines subclass- and light-chain-specific anti-mouse Ig antibodies on a single strip. However, this kit was unable to determine the IgG subclass for the mouse monoclonal antibody of clone BAM-CCMV-29-81 (Figure 4A). Therefore, we employed a modular testing system capable of distinguishing between IgG1, IgG2a, IgG2b, IgG2c, and IgG3 subclasses. This system successfully confirmed the IgG2c subclass for both the Protein-A-purified monoclonal antibody (Figure 4B) and hybridoma supernatant (Appendix A) of clone BAM-CCMV-29-81 in agreement with the determination at the DNA level. Both tests also confirmed the presence of a ĸ-light chain in the samples.

### 3.4. Sequencing of Constant Antibody Chain

After determining the sequence of the variable domain and subclass of the antibody, we examined the constant domains of the antibody in more detail. For this purpose, the previously described method [21] for reverse transcription of mRNA into cDNA and subsequent PCR amplification for sequencing of constant regions was adapted (Figure 5A). In detail, the AOligo(dT)-ISPCR-Primer is anchored to the 5′ end of the poly(A)-tail of the mRNA and enables universal PCR amplification due to its complementary sequence to the ISPCR primer. Moreover, CDR3-specific V-(D-)J-primers were designed according to the results of variable region sequencing (Table 1) and were used to amplify the remaining full-length constant regions from the mouse monoclonal antibody of clone BAM-CCMV-29-81. The generated amplicons were approximately 1300 bp for the constant heavy chain and 600 bp for the constant ĸ-light chain (Figure 3B). After cloning into the vector, again, eight plasmids isolated from colony PCR-positive clones containing constant regions of each chain were sent for Sanger sequencing.

The aligned results showed that the insert sequences had been correctly cloned for the antibody heavy and ĸ-light chain constant region in all replicates. Regarding the constant heavy chain, a neighboring guanine and thymine dinucleotide swap (TG to GT) was identified in the C_H_3 region of all sequenced replicates when aligned to the C57BL/6J reference sequence (Figure 5B). This results in an amino acid exchange at position 435 from valine to glycine. Nevertheless, when the C_H_3 sequence of the NOD strain is used for the evaluation, a 100% match with the reference sequence can be shown. Furthermore, no additional discrepancies were identified in the other constant heavy chain regions when compared to the IMGT reference sequences (Appendix A). Likewise, the sequencing results of the ĸ-light chain were 100% identical across multiple clones and with reference sequences from the IMGT database (Appendix A).

Finally, we validated the full-length amino acid sequence of monoclonal antibody BAM-CCMV-29-81 generated by our workflow (Appendix A) by comparison with RNA Illumina sequencing results from Absolute Antibody Ltd., which served as the reference method. The Sanger-derived heavy and light chain sequences exhibited 100% alignment accuracy with the RNA Illumina sequencing results (Appendix A).

### 3.5. CDR Confirmation at the Protein Level with MALDI-TOF MS

The CDR domains of the mouse monoclonal antibody clone BAM-CCMV-29-81, identified by DNA sequencing (Section 3.3), were verified at the protein level using peptide mass fingerprinting. For this purpose, the Protein A-purified antibody was cleaved using both diluted sulfuric acid and trypsin, as described in earlier studies [34,39]. The resulting MALDI-TOF mass spectra of the partially hydrolyzed and trypsin-digested antibody are shown in Figure 6. Measured peptide masses were compared to theoretical values derived from in silico cleavage of the BAM-CCMV-29-81 heavy and light chain sequences. Overall, we achieved 53.7% sequence coverage for the full-length antibody. In total, five out of six CDRs were successfully assigned to peptides generated by either acidic cleavage or tryptic digestion of the antibody. These sequences are identical to the previously determined sequences at the DNA level.

## 4. Discussion

Knowledge of monoclonal antibody sequences at both the DNA and protein levels is a cornerstone of antibody characterization and enables the development of recombinant antibodies as reliable alternatives to hybridoma-derived antibodies [42]. Access to this sequence information helps overcome common issues of quality and specificity often associated with hybridoma-derived antibodies. A study by Bradbury and colleagues revealed that approximately one-third of 185 analyzed hybridoma clones expressed an additional functional chain [43], highlighting a significant source of potential variability. The current methods for full-length antibody sequencing, such as RNA Illumina or Nanopore [5,6,7,8,9] and *de novo* sequencing [13,14,15,16,17], are expensive, and data evaluation is time-consuming. Recent advances provide powerful alternatives, including full-length single-cell B-cell transcriptomics paired with haplotype-resolved immunoglobulin references [11], fluorescence-activated droplet sequencing that directly reads rare hit droplets and recovers paired chains without physical sorting [12], and integrated repertoire, proteomics, and artificial intelligent pipelines [18], but these approaches are not yet practical for routine hybridoma quality control. To overcome these limitations, we developed a simple and cost-effective Sanger-based workflow for full-length sequencing of monoclonal antibodies from hybridoma cell lines.

Based on a small number of Sanger DNA sequencing reads, we identified the top-matched V-(D-)J genes encoding the heavy and light antibody chains of clone BAM-CCMV-29-81 (Table 1). Additionally, IgBLAST analysis revealed an aberrant, non-productive κ-light chain. It is well established that many myeloma fusion partners can retain aberrant κ-light chain transcripts containing a frameshift at the start of the fusion region, which results in a premature termination codon—such as those observed in hybridoma cells derived from the MOPC21 myeloma cell line [44] and its subclones P3-X63-Ag8.653 [45], OUR-1 [46], NS-1 [47], or SP2/0 [48]. The GenBank database includes several known non-functional ĸ-light chains (Accession Nr. X05184.1, M35669.1, JF412706.1, and FN422002.1), which are available as machine-readable text files in the Appendix A. Taking these into account during the data analysis phase allows for the rapid identification of aberrant chains by comparing newly sequenced samples to existing references. Moreover, Subas Satish et al. demonstrated that these aberrant chains occur at a low abundance, with less than 1%, by using a Nanopore-based sequencing method [7]. As a result, the likelihood of exclusively sequencing non-functional chains is extremely low. Nonetheless, sequencing multiple cloned variable domains from a given antibody clone minimizes the impact of aberrant or non-functional chains during data evaluation.

In addition to variable region sequencing, we demonstrated that the IgG isotype of the mouse anti-CCMV monoclonal antibody clone BAM-CCMV-29-81 could be determined at the DNA level already during this early step of the workflow (Figure 4). This approach eliminates the need for commercially available lateral flow assays for isotype determination. It is particularly advantageous when working with mouse antibodies that belong to less common IgG subclasses, such as IgG2c, which may not be reliably detected by all available test systems.

Providing full-length sequence information for monoclonal antibodies derived from hybridoma cells is a critical quality measure to ensure reproducibility of research results across different laboratories [28,29]. To address this, we introduced a second step into the workflow by designing V-(D-)J–specific primers targeting the CDR3 domains. This simple addition enabled the specific amplification and subsequent sequencing of the constant regions of both the heavy and κ-light chains.

In this context, we observed a sequence divergence from the IgG2c heavy chain gene reference of the C57BL/6J strain (Figure 4) and verified that the sequence matched the IgG2c heavy chain gene from the NOD mouse strain. This highlights the importance of full-length sequencing, as the amino acid sequences of the constant regions of IgG2c heavy chains from C57BL/6J and NOD strains differ by only 0.3% [35,37].

Moreover, by assembling the sequenced variable and constant regions, the complete monoclonal antibody sequence can be conveniently stored in silico or on plasmids, minimizing reliance on the original hybridoma clone and avoiding potential productivity loss. Therefore, this extended protocol also facilitates straightforward in-house generation of expression plasmids for recombinant antibody production.

In this study, we further demonstrated that the predicted CDRs identified through IgBLAST analysis of DNA sequences correspond to peptides detected experimentally by MALDI-TOF MS (Figure 6). The applied fingerprinting method, which can be completed in under one hour, enables rapid verification of the identity of important CDR domains at the protein level. Furthermore, Tscheuschner and colleagues demonstrated that isotype determination of antibodies is also feasible using this fingerprinting method by simply comparing in silico-generated peptides of known constant C_H_1–C_H_3 sequences to the measured peptides in the fingerprint spectrum. Their online, open-source software ABID automatically matches these peptides and suggests the most likely subclass of the analyzed antibody [39]. However, this was not possible in this study, as IgG2c antibodies occur extremely rarely, and therefore, no reference spectra were available for the evaluation. Where deeper repertoire questions arise, for example, clonal selection, isotype switching, or allele-specific usage, single-cell long-read data sets can provide added resolution, although at higher cost and complexity [11].

Furthermore, we showed 100% identity between the antibody sequence obtained in-house via Sanger sequencing and results generated through RNA Illumina sequencing by an external provider. This confirms that costly full-length sequencing services and the risky shipment of hybridoma cells are unnecessary when using the presented protocol. In addition, Sanger sequencing data is significantly easier to interpret than the large datasets produced by RNA Illumina or Nanopore sequencing platforms.

Using this method, we obtained the full-length sequence of a monoclonal antibody in less than 10 working days, making it a time-efficient approach for hybridoma sequencing when high throughput is not required. Parallel processing of multiple antibody clones can further accelerate the workflow. The low equipment and expertise requirements make this method easily adaptable in any molecular biology laboratory. An additional advantage is that the sequenced gene segments are directly available on plasmids, ready for subcloning into expression vectors for recombinant full-length antibody production. This eliminates the need for synthetic gene construction based on RNA-seq data. Moreover, the protocol is readily transferable to the full-length sequencing of human monoclonal antibodies or those from other species.

## Figures and Tables

**Figure 1 antibodies-14-00072-f001:**
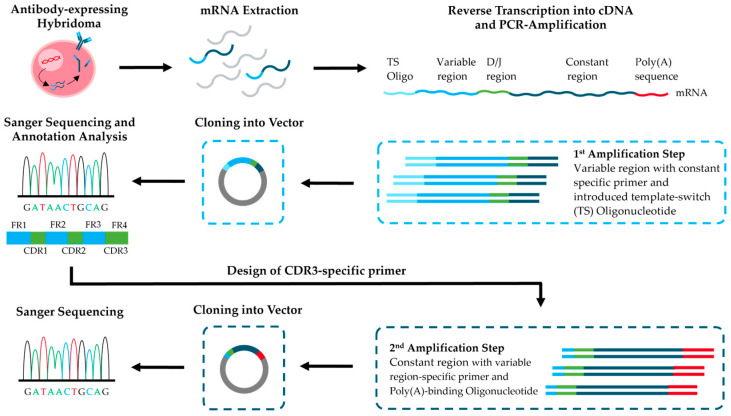
Schematic overview of the two-step workflow for full-length sequencing of hybridoma-derived monoclonal antibodies.

**Figure 2 antibodies-14-00072-f002:**
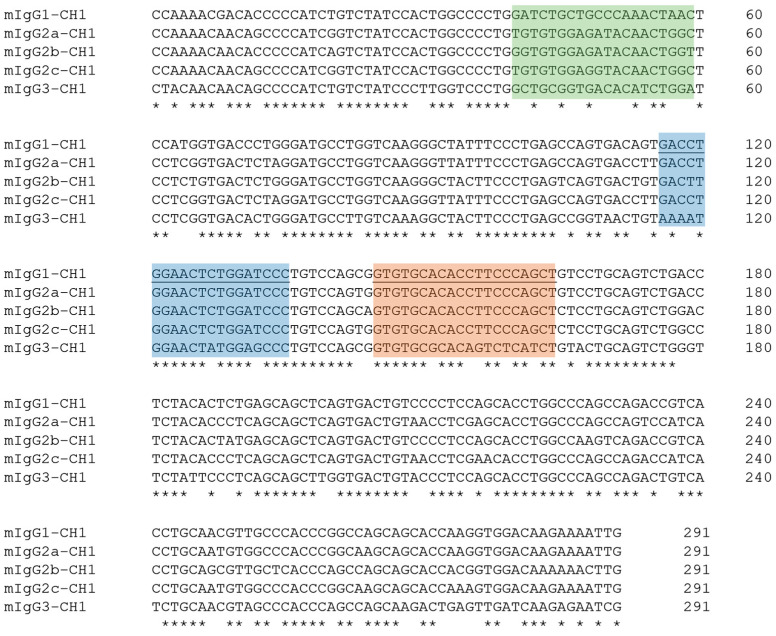
DNA alignment from the C_H_1 exon region of all mouse IgG subclasses. Bases are marked with an asterisk if they are conserved in all subclasses. Binding sites of primers for reverse transcription and PCR amplification are marked in orange and blue, respectively. The underlined sequence represents the primer complement site. The sequence motifs with the greatest differences suitable for subtype determination are highlighted in green. The full nucleotide sequences of reference sequences are available as machine-readable text files in the Appendix A.

**Figure 3 antibodies-14-00072-f003:**
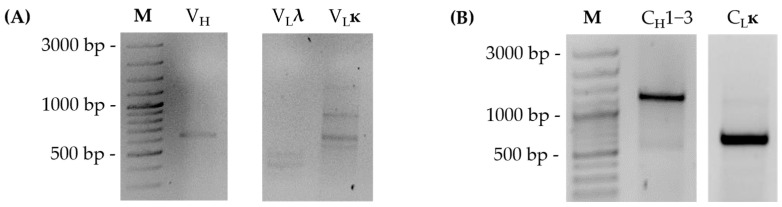
RT-PCR amplification of antibody variable regions with chain-specific primers (**A**) and antibody constant regions with CDR3-specific and AOligo(dT)-ISPCR primers (**B**). ĸ = kappa chain; λ = lambda chain; V_H_ = variable heavy chain; C_H_1-3 = constant domains heavy chain; V_L_ = variable light chain; C_L_ = constant domains light chain.

**Figure 4 antibodies-14-00072-f004:**
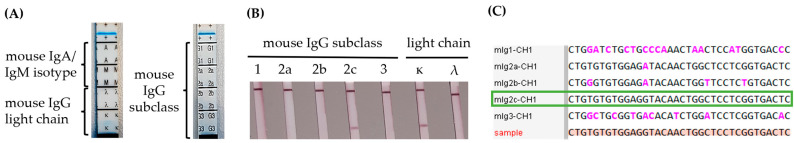
Determination of IgG subclass from Protein A-purified mouse monoclonal antibody of clone BAM-CCMV-29-81 with the Roche IsoStrip (**A**) and the Milenia Biotec modular LFA (**B**) compared to subclass identification on DNA level (**C**). The nucleotides marked in pink show deviations from the DNA sequence of the sample. The red sequence illustrates the sequenced sample, while the green bordered sequence indicates the corresponding reference. The full nucleotide sequences of reference sequences are available as machine-readable text files in the Appendix A.

**Figure 5 antibodies-14-00072-f005:**
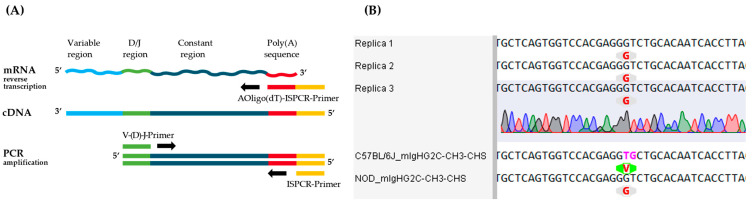
Investigation of constant regions from mouse monoclonal antibody of clone BAM-CCMV-29-81. (**A**) Schematic workflow for cDNA synthesis with AOligo(dT)-ISPCR-Primer and specific PCR amplification of constant antibody region with V-(D-)J-primer. (**B**) Sequence section including differences in the C_H_3 region of the C57BL/6J and NOD strain reference sequences in comparison to multiple sequenced clones of BAM-CCMV-29-81. Pink-marked nucleotides in the C57BL/6J reference differ from the replicate sequences, resulting in a codon encoding valine (V, green) instead of glycine (G, grey). The full nucleotide sequences are available as machine-readable text files in the Appendix A.

**Figure 6 antibodies-14-00072-f006:**
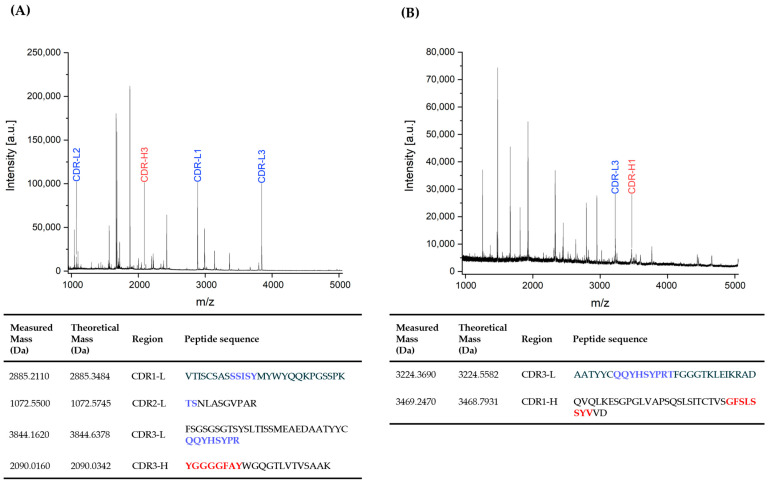
MALDI-TOF fingerprint spectra of mouse monoclonal antibody of clone BAM-CCMV-29-81 after trypsin cleavage for 15 min at 55 °C (**A**) and partial acidic hydrolysis for 30 min at 99 °C (**B**). Colored annotations mark peptides derived from CDRs. The corresponding peptide sequences containing CDRs are listed below. The heavy chain (H) and light chain (L) CDRs 1 to 3 are highlighted in red and blue.

**Table 1 antibodies-14-00072-t001:** Overview of V-(D)-J rearrangement for query sequences of the variable region of the heavy (V_H_) and ĸ-light (V_L_K) chains from a mouse monoclonal antibody of clone BAM-CCMV-29-81. The number after the asterisk indicates the specific allele of the genes.

Chain Type	Top Gene Match	Total Identity (%) to IgBLAST	CDR3 Nucleotide Sequence
V_H_	IGHV-2*01, IGHV2-6-8*01, IGHD-1*01, IGHD-1*02, IGHJ3*01	98.6	5′-GTCAGATACGGTGGTGGAGGGTTTGCTTAC-3′
V_L_ĸ	IGKV4-61*01, IGKJ*01	99.3	5′-CAGCAGTATCATAGTTACCCACGGACG-3′

## Data Availability

The sequence data presented in the article is available in machine-readable text files in the Appendix A. Sequence data of the mouse anti-CCMV monoclonal antibody clone BAM-CCMV-29-81 are available in the GenBank database under accession numbers [PX123808] (heavy chain) and [PX123807] (ĸ-light chain).

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
