# Peer review of "Cost-Effective Method for Full-Length Sequencing of Monoclonal Antibodies from Hybridoma Cells"

_2073-4468, 2025, doi:10.3390/antib14030072_

Round 1

Reviewer 1 Report

Comments and Suggestions for Authors

In this manuscript, the authors described new or improved method for full-length sequencing of antibody. It also could be useful to obtain the DNA sequence of the antibody for recombinant expression. The results are pioneers and I recommend the manuscript be published in “Antibodies” after minor improvements for presentation in followed points:

The manuscript submitted in 2025, but no any references for 2025 and 2024, only 3 for 2023. It means that references are not “fresh”. It will better to add additional references deals with full-length sequencing of monoclonal antibodies. For example, searching “full-length sequencing of antibodies” will give like these papers.

Juncheng Wang, Meng Liu, Rukhshan Zahid, Wenjie Zhang, Zecheng Cai, Yan Liang, Die Li, Jiasheng Hao, Yuekang Xu. Identification of Two Critical Contact Residues in a Pathogenic Epitope from Tetranectin for Monoclonal Antibody Binding and Preparation of Single-Chain Variable Fragments. Biomolecules 2025, 15(8), 1100; https://doi.org/10.3390/biom15081100

Maria Rubtsova, Yuliana Mokrushina, Dmitry Andreev, Maria Poteshnova, Nikita Shepelev, Mariya Koryagina, Ekaterina Moiseeva, Diana Malabuiok, Yury Prokopenko, Stanislav Terekhov, Aleksander Chernov, Elena Vodovozova, Ivan Smirnov, Olga Dontsova, Alexander Gabibov, Yury Rubtsov. A Luciferase-Based Approach for Functional Screening of 5′ and 3′ Untranslated Regions of the mRNA Component for mRNA Vaccines. Vaccines 2025, 13(5), 530; https://doi.org/10.3390/vaccines13050530

Valentina S. Nesmeyanova, Daniil V. Shanshin, Denis E. Murashkin, Dmitriy N. Shcherbakov. Construction of an Integration Vector with a Chimeric Signal Peptide for the Expression of Monoclonal Antibodies in Mammalian Cells. Curr. Issues Mol. Biol. 2024, 46(12), 14464-14475; https://doi.org/10.3390/cimb46120868

These paper could be consider added to manuscript and the differences and novelty must be described.

Gaby Bosc-Bierne, Michael G. Weller. Investigation of Impurities in Peptide Pools. Separations 2025, 12(2), 36; https://doi.org/10.3390/separations12020036

Sarah Döring, Michael G. Weller, Yvonne Reinders, Zoltán Konthur, Carsten Jaeger. Challenges and Insights in Absolute Quantification of Recombinant Therapeutic Antibodies by Mass Spectrometry: An Introductory Review. Antibodies 2025, 14(1), 3; https://doi.org/10.3390/antib14010003

The quality for Fig 5 and Fig 6 must be improved

In supplementary materials Figure S1 could OK, but Fig S2, S3 and S4 are bad quality. It could be improved.

Language is not bad, but it is too complicate to read. Maybe some modification and description for results must be in more simple way, which could be easy understand for non-specialist in this area research.

Author Response

Comments 1: The manuscript submitted in 2025, but no any references for 2025 and 2024, only 3 for 2023. It means that references are not “fresh”. It will better to add additional references deals with full-length sequencing of monoclonal antibodies. For example, searching “full-length sequencing of antibodies” will give like these papers.

Response 1: Thank you for pointing this out. We agree with this comment and have added recent 2024–2025 references in both the Introduction (p. 2) and the Discussion (pp. 12–13).

Added to the Introduction (p. 2):

“More recently, full-length single-cell B cell transcriptomics paired with haplotype-resolved germline immunoglobulin references can read complete heavy and light chain transcripts and assign isotypes unambiguously [11], but these platforms remain specialized and computationally intensive for routine use. In parallel, fluorescence-activated droplet sequencing can directly sequence rare hit droplets and recover paired chains without physical sorting [12], although it requires dedicated microfluidic hardware.”

“Comprehensive reviews highlight a broader trend toward integrating single-cell sequencing, proteomics and artificial intelligence for discovery, but with substantial cost and complexity for everyday laboratory workflows [18].”

            References added:

[11] Beaulaurier, J.; Ly, L.; Duty, J.A.; Tyer, C.; Stevens, C.; Hung, C.-T.; Sookdeo, A.; Drong, A.W.; Kowdle, S.; Turner, D.J. De novo antibody discovery in human blood from full-length single B cell transcriptomics and matching haplotyped-resolved germline assemblies. Genome Res. 2025, 35, 929–941, doi:10.1101/gr.279392.124.

[12] Autour, A.; Merten, C.A. Fluorescence-activated droplet sequencing (FAD-seq) directly provides sequences of screening hits in antibody discovery. Proceedings of the National Academy of Sciences 2024, 121, e2405342121, doi:10.1073/pnas.2405342121.

[18] Townsend, D.R.; Towers, D.M.; Lavinder, J.J.; Ippolito, G.C. Innovations and trends in antibody repertoire analysis. Current Opinion in Biotechnology 2024, 86, 103082, doi:10.1016/j.copbio.2024.103082.

Added to the Discussion (pp. 12–13):

“Recent advances provide powerful alternatives, including full-length single-cell B-cell transcriptomics paired with haplotype-resolved immunoglobulin references [11], fluorescence-activated droplet sequencing that directly reads rare hit droplets and recovers paired chains without physical sorting [12], and integrated repertoire, proteomics and artificial intelligent pipelines [18], but these approaches are not yet practical for routine hybridoma quality control.”

“Where deeper repertoire questions arise, for example clonal selection, isotype switching or allele-specific usage, single-cell long-read data sets can provide added resolution, although at higher cost and complexity [11].”

Comments 2: The quality for Fig 5 and Fig 6 must be improved. In supplementary materials Figure S1 could OK, but Fig S2, S3 and S4 are bad quality. It could be improved.

Response 2: Thank you for this suggestion. We agree with this comment and replace all figures in the manuscript and supplementary materials with high-resolution versions.

Comments 3: Language is not bad, but it is too complicate to read. Maybe some modification and description for results must be in more simple way, which could be easy understand for non-specialist in this area research.

Response 3: Thank you for this comment. We tried to describe some of the results in all sections in an easier and more understandable way and shortened long sentences.

Example Section 3.1, page 7:

“Regarding the κ-light chain, a single conserved constant chain exists which serves as the target for binding by the primers for reverse transcription and amplification, with 100 % consistency (data not shown).” (old version)

“For the κ-light chain, a single conserved constant region serves as the primer-binding site for reverse transcription and PCR amplification. The primer sequences show 100% identity to this region (data not shown).” (new version)

Reviewer 2 Report

Comments and Suggestions for Authors

The authors describe the PCR-based full-length sequencing of a single monoclonal antibody and compare their results with RNA sequencing and MS-based methods. They argue that the described approach is cheap, fast and can help overcome monoclonal antibody reagent reproducibility issues by obtaining and storing sequence information.

Various variants of PCR-based VH-VL region cloning have been in use for three decades and while the description is thorough and detailed it does not introduce relevant novel ideas. If executed properly, all sequencing approaches should yield identical sequences for a given monoclonal antibody. Nevertheless, it is a correct compilation and comparison of the methods that can be used for isotyping and sequencing.

Comments

The authors argue that knowledge of the constant region sequences is important for correct monoclonal identification. Since this is an important message of their article it would be useful to mention and shortly explain with references whether and how the constant regions can influence affinity or specificity of the binding.

The reviewer suggests that the identified sequence be submitted to a nucleotide or protein database.

Author Response

Comments 1: The authors describe the PCR-based full-length sequencing of a single monoclonal antibody and compare their results with RNA sequencing and MS-based methods. They argue that the described approach is cheap, fast and can help overcome monoclonal antibody reagent reproducibility issues by obtaining and storing sequence information. Various variants of PCR-based VH-VL region cloning have been in use for three decades and while the description is thorough and detailed it does not introduce relevant novel ideas. If executed properly, all sequencing approaches should yield identical sequences for a given monoclonal antibody. Nevertheless, it is a correct compilation and comparison of the methods that can be used for isotyping and sequencing.

Response 1: We thank the reviewer for the thoughtful assessment. We agree that PCR-based VH/VL cloning is established. Our contribution is a practical integration tailored for routine hybridoma QC. To avoid implying methodological novelty beyond this scope, we clarified the practical additions in the Introduction, final paragraph:

  • we determine the IgG subclass at the DNA level from CH1 with clear IgG2a versus IgG2c discrimination;
  • we link V(D)J to the constant region in one step using CDR3-anchored primers to obtain full-length, plasmid-ready sequences;
  • we confirm CDR identity at the protein level by a rapid MALDI-TOF peptide fingerprint;
  • we cross-validate both chains against external RNA-Illumina data with 100% identity and document an aberrant κ

The workflow uses standard equipment, completes in <2 weeks, and reduces cost and complexity relative to RNA-seq or de novo MS methods.

Comments 2: The authors argue that knowledge of the constant region sequences is important for correct monoclonal identification. Since this is an important message of their article it would be useful to mention and shortly explain with references whether and how the constant regions can influence affinity or specificity of the binding.

Response 2: We welcome the suggestion on pointing out the influence affinity or specificity of the antibody binding in introduction, page 2.  

“The IgG constant region can also influence how strongly and specifically an antibody binds to its target [27]. Differences in the constant region, for example between subclasses, can change the position or flexibility of the antigen-binding sites and thereby alter binding properties.”

Reference added:

[27] Torres, M.; Casadevall, A. The immunoglobulin constant region contributes to affinity and specificity. Trends in immunology 2008, 29, 91–97, doi:10.1016/j.it.2007.11.004

Comments 3: The reviewer suggests that the identified sequence be submitted to a nucleotide or protein database.

Response 3: Thank you for this valuable suggestion. We have included the full nucleotide and amino acid sequences in the Supplementary Material (Table S4) and deposited both chains in NCBI GenBank under the accession numbers PX123807 (kappa chain of the anti-CCMV antibody) and PX123808 (heavy chain of the anti-CCMV antibody). The accession numbers have been added to the Materials and Methods section (2.5 Vector cloning, Sanger sequencing and data evaluation, page 6) and to the Data Availability Statement (page 14).